# Consumers’ Attitude towards Sustainability in Italy: Process of Validation of a Duly Designed Questionnaire

**DOI:** 10.3390/foods11172629

**Published:** 2022-08-30

**Authors:** Vittoria Aureli, Alessandra Nardi, Daniele Peluso, Umberto Scognamiglio, Laura Rossi

**Affiliations:** 1CREA Council for Agricultural Research and Economics-Research Centre for Food and Nutrition, 00178 Rome, Italy; 2Department of Mathematics, University of Rome “Tor Vergata”, 00133 Rome, Italy; 3Bioinformatics e Biostatistics Unit, Istituto di Ricovero e Cura a Carattere Scientifico (IRCCS) Fondazione Santa Lucia, 00179 Rome, Italy

**Keywords:** validation, consumers, sustainability

## Abstract

This study aimed to describe the process of validation of a questionnaire assessing Italian consumers’ perception of food sustainability. The study has a multiphase design. Phase 1 consisted in translating and structuring the questionnaire. Phase 2 aimed at assessing the validity of the content by experts. Phase 3 consisted of a pilot study (n = 150) carried out to revise the questionnaire based on the reactions of consumers representing the target group of the assessment. The questionnaire showed adequate content validity for 11 out of 14 questions (>0.79) and S-CVI/Ave > 0.80. Cronbach’s alpha values ranged from 0.08 to 0.90. The construct with insufficient results (0.08) was changed because it failed to correlate with the rest of the questionnaire. The factor analysis permitted the identification of questions that needed improvement in terms of comprehensibility, elimination of redundancies, and repetitions. The validated questionnaire included 12 questions (71 response options); 3 sections were identified: food sustainability knowledge (4 questions-30 items); sources of proteins alternative to meat (3 questions-20 items); eating behaviors (5 questions-21 items). This study showed the importance of validation before the administration on a large scale of a questionnaire on a topic such as sustainability still lacking large support from consensus documents.

## 1. Introduction

According to the Food and Agriculture Organization (FAO) and the World Health Organization (WHO), Sustainable Healthy Diets are dietary patterns that promote all dimensions of individuals’ health and wellbeing; have low environmental pressure and impact; are accessible, affordable, safe, and equitable, and are culturally acceptable [1,2]. In recent years, several studies highlighted the negative impact that global food production has on water consumption, land use, and gas production. Considering the detrimental impact of current food systems, and the concerns raised about their sustainability, there is the need to promote diets that are protective of human health and the environment. Animal products significantly contribute to greenhouse gas emissions (GHGEs) responsible for climate change [3]. Among different animal foods, meat is the most impacting in terms of land use and water consumption [4]. As reported by Nelson et al. [5] a dietary pattern rich in plant foods, such as vegetables, fruits, whole grains, legumes, nuts, and seeds, and low in animal foods is the most protective of the environment.

Shift toward more sustainable food patterns may be relevant not only for the environment perspective but also for public health consequences [6,7,8].

It is of primary importance to know how sustainability is considered by the consumers and what practicable and acceptable food changes could be proposed. Consumers can largely contribute to environmental protection, preferring foods with high nutritional value and low environmental impact (few animal products and most plant-based foods) [9]. These choices, however, are often conditioned by the level of knowledge, as resulting from a study carried out by Peschel et al. [10], further confirmed by Hartmann et al. (2021), reporting that the consumers’ knowledge is an important driver of motivation for appropriate food choices and habits environmentally protective [11].

Consumer attitude toward sustainability was studied, among others, with tools analyzing food waste and diet quality [12], knowledge of the environmental impact of food [11], and consumer literacy [13]. Despite the increasing evidence of the huge impact on the ecosystems of consumers’ dietary habits, most people are not aware of the environmental effects of food production and consumption [11]. In addition to that, the sustainability of diet is an aspect still not completely exploited, in which the risk of bias and “personal” interpretation is possible also in consideration of the limited sources of information based on consensus documents. The composition of diets and the quality of foods have direct effects on human health, a well-known and consolidated concept. However, the indirect health effects caused by environmental changes associated with the processes of producing foods are less recognized. National dietary guidelines aim to provide advice for constructing healthy diets, thus the guidelines should consider both direct and indirect health consequences of the nutritional recommendations, including the environmental implications of food choices [14]. In the light of this commitment, several countries have started to incorporate sustainability considerations into their food policies and consumer education programs. Given the policy and programmatic implications of dietary guidelines, the development and integration of recommendations that promote specific food practices and choices would represent an obvious strategy for addressing sustainability, mainly in its nutrition and environmental dimensions. Such recommendations include for example: having a mostly plant-based diet, focusing on seasonal and local foods, reduction of food waste, consumption of fish from sustainable stocks, and reduction of red and processed meat, highly processed foods, and sugar-sweetened beverages [15]. At the European Level guidance on sustainability were mainly provided in term of recommendations for the selection of local seasonal products and reducing waste; however, frequently these advices did not have an explicit mention of environmental aspects [16]. In the newly released Italian Dietary Guidelines [17], practical suggestions aimed to improve consumer’s behaviors in terms of environmental protection were provided in addition to health-promoting recommendations. One of the limitations of the Italian approach of the development of sustainability recommendations was that it was carried out without an evaluation of Italian consumers’ awareness of sustainability.

The purpose of this study is to develop a validated tool for assessing the consumers’ perception and understanding of sustainability. To this scope, we started with something already existing and tested getting inspiration from the experience carried out by The European Consumer Organization (*BEUC-Bureau Européen des Unions de Consommateurs*) that assessed consumers’ attitudes toward food sustainability as well as their awareness of the impact of food choices on the environment. The survey was conducted simultaneously in 11 European Countries (Austria, Belgium, Germany, Greece, Italy, Lithuania, the Netherlands, Portugal, Slovakia, Slovenia, and Spain) in the period October-November 2019. The questionnaire was administered to 1000 respondents per country selected on the basis of quotas for age, sex, and geographical region (established on consideration of the distribution of the general national population) [18].

As a first step of the work, the mentioned questionnaire was translated and adapted to the Italian context. Approaching the themes of sustainability, we consider it important also to explore the topic of the source of proteins alternative to meat, a subject largely debated at the European Union level [19,20] that was also a matter of several calls for innovative research projects [21] as well as of studies on consumers’ acceptability [22,23,24].

Considering all the aspects that were changed and added to the already tested questionnaire, we considered it important to perform a process of validation of the newly developed data collection tool before using the instrument on a large scale. The main aim of this paper is to describe the process of validation showing the rationale of the changes carried out to the original questionary that led to the production of a reliable instrument to be used for national survey and, potentially, for other kinds of assessment. Specifically, the questionnaire resulting from the validation process aims to evaluate consumers’ attitudes toward the themes of sustainability with a consolidated scientific methodology reinforcing and providing value to the findings of future assessments. A methodological improvement of the originally developed questionnaire [18] is carried out and the extent of the changes after the validation process is shown. The research questions underlying this work are: (i) what is the level of adaptation of a questionnaire developed in an international (European) context once applied at the national level? (ii) what lessons could be capitalized from this process of adaptation and validation? (iii) considering the increasing popularity of online assessments what recommendations could be provided for other studies, in particular those involving emerging topics?

## 2. Materials and Methods

### 2.1. Design of the Study

The questionnaire as originally proposed (Appendix A) and the questionnaire resulting from the validation process (Appendix A) could be found in the Appendix A.

The questionnaire validation protocol was designed following a procedure largely reported in the literature [25,26,27,28], applied in different settings, and assessing different topics and constructs. The procedure could be summarized as follow: the initial analysis for the identification of the domains of interest was normally carried out by the team that developed the questionnaire [29], then the validity of the items was assessed in consecutive stages by requesting individual experts to evaluate the content relevance and simplicity of individual items and the entire set of items (questionnaire) as a tool [30,31,32]. Finally, the questionnaire was pilot tested in a sample of the target group. Hence, the stepwise procedure of the validation process has been divided into three phases (Figure 1):Phase 1: Questionnaire translation, construction, and content formal review;Phase 2: Content validity;Phase 3: Pilot study.

#### 2.1.1. Phase 1: Instrument Translation, Construction, and Content Formal Review

In the first step of phase 1, the original questionnaire of the European Consumer Organisation [18] was translated into Italian. The translation was carried out by the authors of the present paper that were aware of the concepts that the questionnaire intended to measure and would provide a translation as much as possible closest to the original instrument [33]. No difficulties in the translation were encountered since the original questionnaire included items written in simple, short, and familiar language adapted to be disseminated among consumers. Considering the absence of misunderstandings or unclear wordings in the initial translations, the back-translation of the questionnaire into the original language was not carried out. The original questionnaire [18] was created to better understand attitudes toward sustainable food consumption and the extent to which consumers realize the impact that their food choices have on the environment. The obstacles faced by consumers in adopting more sustainable food habits, and the measures they think are needed to make the sustainable choice easier were included in the assessment. In addition to that, for Italian consumers, it was considered relevant to assess the attitude toward sources of proteins alternative to meat. To this scope, a narrative review of the scientific literature of the most common repositories (e.g., Pubmed, Google Scholar, etc.,) was carried out to search for questionnaires measuring the attitude of consumers toward the consumption of meat and the alternative sources of proteins. The formal review of the content was carried out by sending a preliminary version of the questionnaire to a first expert group including nine academic scientists with food and nutrition background to get general feedback on the structure and content of the tool. In an email explaining the aim of the study, the experts were asked to report any critical issue raised during the compilation.

#### 2.1.2. Phase 2: Content Validity

In phase 2, the questionnaire content validity was carried out using the methodology reported by Rodrigues et al. [34], Caruso et al. [35], and Yusoff [30]. The content validity process consisted in the revision of the questionnaire by a second group of 10 experts, different from those of the phase 1, having competencies in nutrition and public health and coming from academia and public research institutions. Instructions on how to evaluate each item were sent with a cover letter explaining the procedure and the reasons for the content validity process. Experts were required to assess importance, clarity and necessity for each item with a Likert scale values from 1 to 4 (1 = Not relevant, 2 = Somehow relevant, 3 = Quite relevant, 4 = Relevant). For the data analysis, the values 1 to 4 were converted into dichotomous values (0/1) with 0 corresponding to 1 and 2 and 1 corresponding to 3 and 4. As reported in different studies [30,36] item validity was determined by calculating the content validity index for each item (I-CVI) and the scale-level content validity index (S-CVI). To evaluate the content validity of the questionnaire the following indices were used:I-CVI (item-level content validity index)Formula: *I-CVI = (agreed item)/(number of experts)*S-CVI/Ave (scale-level content validity index based on the average method)Formula (1): *S-CVI/Ave (based on I-CVI) = (sum of I-CVI scores)/(number of items)*Formula (2): *S-CVI/Ave (based on proportion relevance) = (sum of proportion relevance rating)/(number of experts)*S-CVI/UA (scale-level content validity index based on the universal agreement method)Formula: *S-CVI/UA = (sum of UA scores)/(number of items)*
where UA corresponds to Universal Agreement: the score “1” was assigned only to the questions that received 100% relevance from all experts; other questions received a “0” score.

#### 2.1.3. Phase 3: Pilot Study

The questionnaire resulting from phases 1 and 2 was distributed through social media channels (WhatsApp, Facebook, Instagram, Linkedin) for 4 weeks (September 2021) with the finality of reaching a sample of 150 adults (>18 years). Applying the sampling reported by other studies [37,38], the pilot test was carried out using convenient subjects’ inclusion without setting specific exclusion criteria. This simple non-probabilistic methodology to recruiting respondents online, by inviting them to follow a link to a survey placed on a web page, email, or other similar means, is defined as “river” sampling by Lehdonvirta et al. [39]. A set of questions covering socio-demographic, personal information, and professional characteristics of the participants was included, asking for sex, age, weight (kg), height (cm), level of education, family composition, city of residence, type of work, and average annual income.

### 2.2. Statistical Analysis

Continuous items were reported as median and quartiles; absolute frequencies and percentages were used to describe categorical items. Reliability analysis was undertaken to evaluate the quality and the overall consistency of the questionnaire. Internal consistency of the questions’ blocks was measured by using Cronbach’s alpha coefficient. For Cronbach’s calculation, negative wording questions were reversed into a positive scale. Explanatory factor analysis was performed to determine the construct validity of the questionnaire. The principal component analysis method was used for the extraction of factors. Varimax rotation was applied to optimize the loading factor of each item.

For all the extracted factors, eigenvalues, proportion, and cumulative proportions of explained common variance were computed. Then, factors with a proportion of explained common variance greater than 3% were selected. For each item, communality, maximum factor loading, and corresponding factor were reported.

## 3. Results

### 3.1. Questionnaire Construction and Content Formal Review (First Expert Group)

As a result of the scientific literature analysis aimed to search for questionnaires measuring the attitude of consumers on meat consumption and alternative sources of proteins, the following four questions were added to the original questionnaire [18]:A question on meat substitute (Q8) from Schösler et al. [22]A question on the role of meat in the diet (Q10) from Tarrega et al. [23]A question on consumers’ meat consumption attitude after changing production methods with increasing prices (Q11) and a question on different activities’ contribution to climate changes (Q14) from Malek et al. [24]

After the initial pool of the questionnaire, qualified experts reviewed the items. According to Tsang et al. [33], the purpose of this revision is to check for accuracy, absence of construction problems, and grammatical correctness. The formal review of the content was preceded by a thorough explanation of the role of the experts and the purpose of the revision. The experts were informed about the scope of the work and the importance to provide comments aimed to improve the quality of the assessment. Active and conscious participation of the experts was required. A general consensus of the experts on the relevance and validity of the questionnaire was reached with comments that were mainly related to redundancies and unclear formulations of the items. The text was revised according to the provided inputs and was transcribed on Google Form^®^ to automatize the data collection. The questionnaire resulting from phase 1, available in the Appendix A, undertook the following phases of the validation process.

### 3.2. Content Validity (Second Experts Group)

Table 1 reports the content validity indices calculated from the second experts’ group responses in phase 2 of the validation process. The I-CVI for 11 out of 14 questions exceeds the value of 0.79, which is the threshold for considering the question relevant [34]. For questions Q4, Q9, and Q10 that showed a value of I-CVI less than 0.79 (0.70), a reformulation of the items was carried out to improve the comprehensibility. The calculation of the S-CVI/Ave index based on the average of I-CVI scores across all items divided by the number of items (n = 14), accounted for 0.86 exceeding the cut-off point of 0.80 [32]. However, the calculation of the S-CVI/UA index showed a value of 0.36 lower than the acceptable cut-off point (0.80). This low index was related to the fact that only 4 out of 14 questions were considered with 100% of content validity.

Table 1 reported also the average values obtained from the individual questions using the Likert scale. As shown from the average (SD) column no value falls below 3 (quite relevant), confirming that the experts evaluated the questionnaire as generally acceptable.

### 3.3. Pilot Experiment Outcomes and Relative Changes in the Questionnaire

In the pilot experiment, 150 respondents were recruited. The socio-demographic and personal characteristics of the sample were reported in Appendix A. The majority of respondents (74.6%) were women with a high level of education (74%); approximately one-third (35.3%) of the sample was aged 50–69 years. No polarization of responses with scoring all near the bottom or near the top was observed since items were not eliminated for these reasons at this stage.

#### 3.3.1. General Analysis

The first step of the qualitative analysis was the evaluation and interpretation of the missing answers. Questions with a percentage of missing greater than 10% were considered with limited comprehensibility and were removed or reformulated. A quota of missing answers exceeding the fixed cut-off level was recorded for Q1—response option 3 (18.5%) and Q10—response option 5 (11.3%) that were removed. Questions Q6—response option 6 and Q6—response option 7 obtained 13.3% (average) of missing answers since were further specified and detailed, and Q13—option 8 (15.3% of missing answers) was reformulated.

As a general change, we decided to remove the “*I do not know*” option except for block Q9, where this option could be an answer itself. In addition, the option “*yes, if I like it*” was added to Q9 to differentiate between respondents that would always refuse the consumption of proteins alternative to meat and consumers who would be favorable to consumption in case of taste acceptance.

As a result of the pilot test, the majority of the answers were converted into continuous scales from 1 (very disagree) to 10 (very agree), with the obligation to answer for each item. Further reformulations were carried out for questions Q3, Q5, and Q8, allowing a dichotomous answer (Yes/No) for each item instead of limiting the selection to 3 response options.

#### 3.3.2. Quantitative Analysis

Table 2 shows the Cronbach’s alpha values calculated for each block of items having the Likert scale response options. According to this analysis, the group Q10 resulted very reliable having the highest internal consistency values (0.90). Blocks Q6 and Q13 have an intermediate acceptable value, while items of Q1 are not internally consistent and needed reformulation in consideration of the very low Cronbach’s alpha value (0.08).

Factor analysis was undertaken for continuous variables related to the response of questions Q1, Q6, Q10, and Q13. Questions that had obtained >10% of “missing” were not considered in this analysis, being already revised in the previous steps of the procedure. Figure 2 shows eigenvalues, proportions, and cumulative proportions of common variance explained by the 21 factors resulting from factor analysis. Among them, seven factors were selected, explaining a proportion of the common variance greater than 3%. However, it should be pointed out that, the first four factors accounted for 90% of the common variance.

Table 3 reports factor loading after Varimax rotation, corresponding factor, and communality. According to this analysis, the following modifications to the original questionnaire were carried out. Factor 1 confirmed the homogeneity and acceptability of the items of the Q10 block that was left as in the original version. Factors loadings related to Factor 2 supported the internal coherence of the Q13 block except for Q13—option 4 and Q13—option 7. These items were also characterized by a low level of commonality, suggesting a very specific content not homogenous with the other response options, and then were reformulated to improve comprehensibility. Factor 3 confirmed the appropriateness of the response options of block Q6, except for Q6—response option 8 and Q6—response option 9. Then, Q6—response option 8 was eliminated considering that the concepts were already present in the other items, and Q6—response option 9 was reformulated changing the negative sense to a positive formulation. Block Q1 resulted in not being homogenous, very far from others, and with limited internal coherence. A new Q1 block was created adding to the original items Q1—response options 1 and 2, the questions Q2 and Q4 that were isolated and with the heterogeneity of answers. In addition to that response options, 4 and 8 of Q13 were added to the Q1 block resulting better in line with the topic of this block.

As a final overall revision, a new order of the questions was defined considering the consistency of the subjects of the different blocks. The validated version of the questionnaire contained 12 questions that included a total of 71 response options. Factor analysis showed that the three first factors included most of the items as represented in the three circles of Figure 3. This permitted to identify three sections of the questionnaire: (i) food sustainability knowledge (4 questions accounting for 30 items); (ii) sources of proteins alternative to meat (3 questions accounting for 20 items); (iii) eating behaviors (5 questions accounting 21 items). The final version of the validated questionnaire is reported in Appendix A which could be compared with the original questionnaire reported in Appendix A.

## 4. Discussion

The aim of this study was to show and discuss the validation process and the adaptation to the Italian context of a questionnaire on consumer knowledge and attitudes toward sustainability starting from a questionnaire already used in Europe [18]. Using a procedure largely described in the literature [25,26,27,28], the questionnaire was initially developed by experts and then revised on the basis of the results of a pilot test on consumers. The analysis of the data collected in the pilot phase on a group of 150 respondents, highlighted a series of inconsistencies that were corrected in the final version of the questionnaire. After this, exercise problems were addressed and amendments were made to develop a validated assessment tool as a prerequisite for reliable data collection [40]. The most important aspect of this research is that, to the best of the authors’ knowledge, this is the first study in which the development and testing of a web questionnaire to evaluate the consumers’ attitude on an emerging topic such as sustainability and proteins alternative to meat was deeply described and discussed. In addition to that, the description of the validation process showed the limits of the use of a non-validated instrument. According to Aithal and Aithal [41], the validation of the questionnaire helps researchers in avoiding non-appropriate data collection and analysis and is an important step for having reliable results. The validation process is of particular relevance for emerging topics for which it is important to know if the questions are correctly understandable and interpretable by the respondents. The validation of experimental data is also essential for the replicability of the results and their interpretation, and is a crucial part of systematic research either empirical or experimental [42]. This is the sense of the present methodological paper in which the process of validation is presented and discussed showing the reasons for the changes made to the questionnaire during the process of revision and adaptation to the Italian context. Based on the existing literature regarding content validity for questionnaire development, this study proposed both item-level and scale-level content validity indices (I-CVI and S-CVI/AV) as quantitative indicators of acceptable content validity [32]. The content validity analysis with the experts’ group confirmed the relevance and appropriateness of the questionnaire resulting with indexes above the fixed thresholds. Rewording of the questions was carried out based on the reviewers’ narrative comments that clarified both strengths and potential gaps of the different proposed response options. The number of answers for some questions was increased following the experts’ input of not limiting to the portfolios of response options. Some terms were better specified because they were considered not clear or not sufficiently consolidated or too vague. Examples of these changes were “meat of plant origin” which was changed to “plant-based meat alternatives” and “waste less food at home” in which it was specified how to do.

The pilot test carried out on a casual sample of consumers provided very relevant information on how to change the questionnaire. This is an important aspect to consider because the perspectives and the interpretation of the survey respondents (in this case the consumers) are different from those of the experts. A pivotal aspect addressed during the validation of the questionnaire was related to the *“I do not know”* option. As reported by Ball [43], among the rules which should be followed during the construction of a survey, there is the possibility of offering to the respondents the option *“I do not know*”, to reduce the non-answering level. There is not a general rule on this aspect. According to Krosnick et al. [44], the large selection of *“I don’t know”* is not an index of low reliability and validity of response per se, but depends on the topic of the questionnaire. The elimination of the option *“I don’t know”* during the validation process of the present questionnaire was carried out to discourage the non-responding attitude stimulating the respondent to take a position. We consider this better fitting with the topic of the questionnaire, sustainability, a newly emerging subject in which the selection of a large number of *“I don’t know”* could be a concrete risk. We maintained the possibility of choosing *“I don’t know”*, only for question Q7 of the final version (Appendix A) considering the response option pertinent to the question.

The original questionnaire was proposed with a heterogeneity of answers scales, multiple and dichotomous. To make the questionnaire better usable to consumers we decided to homogenize the various answers scales to facilitate the compilation. Before the validation, questions had the Likert scale with a variable number of options (1 to 10, 1 to 5, 1 to 4). The Likert scales system is a convenient way to measure constructs, initially introduced with a 5-point option; over the years it has been used with different measurement response options from two to eleven points [45] and the scientific literature regarding the advantages and disadvantages of having a low number of options with respect to large ranges is very heterogeneous and not conclusive. As reported by Taherdoost et al. [46], weakness of the Likert scale with more than 5 points is that participants would avoid extreme response categories, resulting in a central trend bias. According to Preston & Colman (2000) the reliability, validity, and discriminating power of the scales with multiple response categories achieved significantly better performance for the 10-point scale or above [47]. In light of these aspects, during the validation process of the present questionnaire, it was decided to have 10-point Likert scales to ensure sufficient variance of answers between respondents [33]. Another advantage of a 10-points scale is the possibility to consider it as a continuous scale, broadening the possibilities of statistical analysis. Questions with continuous scales were mixed with questions with categorical scales with yes/no response options eliminating any possibility of neutral answers. This decision is in line with the idea of pushing the respondents in taking a position on the themes of sustainability.

According to several validation studies [40], Cronbach’s alpha coefficient was used to evaluate the internal consistency of the questionnaire. In the present work, Cronbach’s alpha values ranged from 0.08 to 0.90. For some authors [40,48], an Alpha value equal to or higher than 0.67 is an acceptable value. Values > 0.80 are considered good results [40] or excellent outcomes [49]. In our case, one construct had values above 0.80 and for other two, Alpha values were 0.67 and 0.63. This analysis permitted to identify the question to be changed meaning that with the Alpha value of 0.08; a value demonstrating the limited correlation of the question with the others and with the general structure of the questionnaire.

The exploratory factorial analysis related to quantitative items permitted further fine-tuning of the questionnaire. Factor analysis is a widely used technique in exploring theoretical constructs to determine the questionnaire’s factorial structure. With this analysis, the evaluation of the consistency and the internal coherence of the original question blocks with quantitative items was performed. The factorial analysis has the methodological merits of providing data for the creation of a rigorous conceptual structure [50]. In our questionnaire, we compared results from the factor analysis with the original block structure. This comparison permitted the identification of the items not consistent with the attributed block and to the identification of a block detached from the rest of the questionnaire that was substantially revised. Another important advantage of the factorial analysis was the possibility of the creation of the three sub-sections of the questionnaire that would result extremely useful for the interpretation of the outcomes of future assessments.

This research has limitations and strengths. The main limitation is that the sample for the pilot is a convenience sample. In open web-based surveys, selection bias occurs inevitably [51] also for the effect of the “river sampling” that has the advantage of the facility to reach a large number of participants at limited costs, and the disadvantage of the selection bias. According to Lehdonvirta et al. [39], this kind of sampling can be used to describe certain phenomena and their safest use is in examining members of the sample itself avoiding inferences that could be affected by the selection biases. We performed the analysis of the pilot data in the light of these methodological inputs considering the pilot test an opportunity for the questionnaire developer to know if there is confusion about any items, and to explore any possible improvements of the items [33]. A limit of the present methodological study is that the content validity evaluation is a subjective procedure, and every expert could not represent all the dimensions of the content domain [27]. Another limitation is related to the fact that questions and response items were not randomized. The questionnaire was conceived by nutrition and public health experts that do not have specific competencies in environmental sustainability issues and also this could be considered a limitation of the research. However, in a modern approach, nutritional recommendations should consider the environmental implications of food choices and the development of nutritional recommendations is a task of community nutrition specialists and public health experts. Our research has also strengths. The sample size used in the present study was in accordance with the classic rule of respondent-to-item ratios established by Kline [52] of using 2 to 20 subjects for each item of the questionnaire. Given the variation in the types of the questionnaire being used, there are no absolute rules for the sample size needed to validate a questionnaire [53]. However, we carried out a retrospective power test, calculating the error associated with the estimate of the average response for items from 1 to 10, assuming the maximum variance observed (10.5). A group of 150 guarantees a standard error of 0.26 (with a confidence interval amplitude of 1.05). Concerning the factor analysis there are several studies in the field that require a range of 3 to 10 subjects for each variable [54,55,56]. Other studies, however, state that the minimum sample size depends on the characteristics of the study, in particular on the level of communality of the variables and the level of factor over determination [57,58,59]. In our case this is an exploratory factor analysis and the ratio between the number of subjects (150) and the number of variables (21) is 7, within the indicated ranges. We have fairly high commonalities for most items and the first four factors explain 90% of the variability. Furthermore, we did not have any convergence problems. It is worthless to say that larger samples are always better than smaller samples and that it is recommended that investigators utilize as large a sample size as possible. However, it is difficult to have large sample size in pilot validations study.

The original questionnaire [18], carried out at the EU level and not in the research setting, did not report a validation procedure. In this study, we demonstrated the importance of the validation process as well as the process of adaptation to the local (Italian) context that provided substantial changes and improvements. Our study highlighted the weaknesses and strengths of the original questionnaire, providing a version adequate and coherent for a large scale assessment. In addition to that, the good results in all the validation indexes, and the outcomes of the reliability tests would guarantee a high quality of the assessment tool.

The validated questionnaire resulting from this study will be used for a national-representative survey aimed at evaluating the degree of Italian consumers’ knowledge and understanding of food sustainability and whether the alternative source of proteins of the new generation or not usual in the Italian food culture, could substitute animal proteins in the diet. Italy would represent an interesting case study in the European context in consideration of the fact that Italian cuisine and consumers’ attitude are largely based on the Mediterranean Diet principle [60], a model that has gained fame and honor, being a model that combines the prevention of Non-Communicable Chronic Diseases, longevity, and health with consumers’ acceptability and sustainability of the productive systems [61]. However in the Mediterranean context, the main sources of vegetable proteins are represented by cereals and legumes [62], the latter often not consumed as an alternative to meat. Since it would be of particular relevance to assess the attitude of Italians towards new foods and sources of proteins alternative to meat. In addition to that, the findings of the nationwide assessment could be used as a benchmark for developing specific actionable recommendations considering the limits of the inclusion of sustainability in nutritional advice contributing to the maximization of the capacity of dietary guidelines in creating a healthy food environment. Evaluation of the impact of these recommendations should be carried out in consideration of the gap between real food choices and good intentions either in terms of healthy eating or sustainability.

## Figures and Tables

**Figure 1 foods-11-02629-f001:**
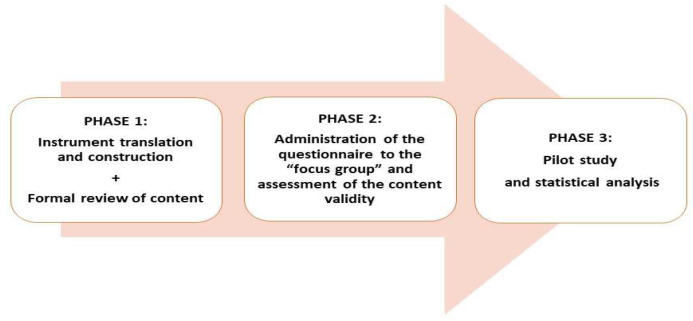
Phases questionnaire validation process.

**Figure 2 foods-11-02629-f002:**
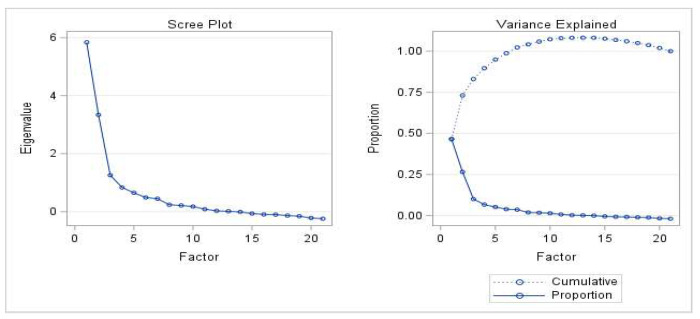
Eigenvalues, proportions, and cumulative proportions of common variance explained by the 21 factors resulting from the factor analysis.

**Figure 3 foods-11-02629-f003:**
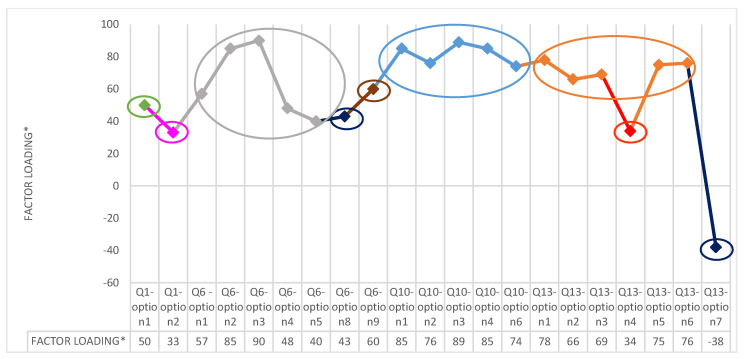
**Figure 3** graphically shows the factor loading data shown in Table 3. The different colors identify the seven factors.

**Table 1 foods-11-02629-t001:** Ratings on the item scale by ten experts.

	Expert 1	Expert 2	Expert 3	Expert 4	Expert 5	Expert 6	Expert 7	Expert 8	Expert 9	Expert 10	Average (SD)	Experts in Agreement	I-CVI	UA
Q 1	1	1	1	0	1	1	0	1	1	1	3.17 (0.21)	8	0.80	0
Q 2	1	1	1	1	1	1	1	1	1	1	3.77 (0.12)	10	1.00	1
Q 3	1	1	1	1	1	1	1	1	1	1	3.80 (0.17)	10	1.00	1
Q 4	1	1	1	0	1	0	1	0	1	1	3.17 (0.06)	7	0.70	0
Q 5	1	1	1	1	1	1	1	1	1	1	3.73 (0.15)	10	1.00	1
Q 6	1	1	1	1	1	0	1	1	1	1	3.50 (0.26)	9	0.90	0
Q 7	1	1	1	1	1	1	1	1	1	1	3.67 (0.06)	10	1.00	1
Q 8	1	1	1	1	1	1	1	1	1	1	3.80 (0)	10	1.00	1
Q 9	1	1	1	1	1	0	0	0	1	1	3.20 (0.35)	7	0.70	0
Q 10	1	1	1	1	1	0	0	0	1	1	3.23 (0.23)	7	0.70	0
Q 11	1	1	1	1	1	0	1	0	1	1	3.23 (0.25)	8	0.80	0
Q 12	1	1	1	1	1	0	0	1	1	1	3.20 (0.36)	8	0.80	0
Q 13	1	1	1	0	1	1	1	1	1	1	3.40 (0.20)	9	0.90	0
Q 14	1	1	0	1	0	1	1	1	1	1	3.50 (0.10)	8	0.80	0
												S-CVI/AVE	0.86	
PR	1.00	1.00	0.93	0.79	0.93	0.57	0.71	0.71	1.00	1.00		S-CVI/UA		0.36
	The average proportion of items judged as relevant across the ten experts						0.86							

**Table 2 foods-11-02629-t002:** Cronbach’s alpha value for each block of items having the Likert scale response option.

	Cronbach’s α
Q1	0.08
Q6	0.67
Q10	0.90
Q13	0.63

**Table 3 foods-11-02629-t003:** Reports factor loadings and communalities after Varimax rotation. * Factor loadings are multiplied by 100 and rounded to the nearest integer. The table with all the factors for each question is presented in Appendix A. The factor loading of each question is shown graphically in Figure 3 identifiable according to the reference colors.

Question	Factor Loading *	Factor	Communality
Q1-option1	50	5	0.30
Q1-option2	33	1	0.41
Q6-option1	57	3	0.63
Q6-option2	85	3	0.81
Q6-option3	90	3	0.82
Q6-option4	48	3	0.72
Q6-option5	40	3	0.47
Q6-option8	43	7	0.28
Q6-option9	60	6	0.40
Q10-option1	85	1	0.86
Q10-option2	76	1	0.69
Q10-option3	89	1	0.85
Q10-option4	85	1	0.77
Q10-option6	74	1	0.58
Q13-option1	78	2	0.88
Q13-option2	66	2	0.63
Q13-option3	69	2	0.72
Q13-option4	34	4	0.23
Q13-option5	75	2	0.70
Q13-option6	76	2	0.86
Q13-option7	−38	7	0.23

## Data Availability

Data is contained within the article or Appendix A.

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
