# Peer review of "Consumers’ Attitude towards Sustainability in Italy: Process of Validation of a Duly Designed Questionnaire"

_foods, 2022, doi:10.3390/foods11172629_

Round 1
Reviewer 1 Report
Foods.1812434.Sustainability.Italy.review
p. 1. The first sentence of the Introduction does not make clear the place of health. Health is included by the FAO (first sentence but not after that)
p. 2 line 45. Health is emphasized in this paragraph, but then it disappears beyond that.
p. 3. The goals of this paper are important: validating national application from an international start. We often see the application of a questionnaire but not often the development of a questionnaire for national usage.
Line 117. Instrument translation. How was this done? With a dictionary? With language experts? With consumers? If only done by experts it could be questioned from a linguistic perspective.
Sections 2.1.1 and 2.1.2. I question the sole use of experts in these roles, with no input by consumers.
Lines 181-186 – please check your English
Line 187. “In the formal review of the content carried out by the first group of experts, comments were mainly related to redundancies and unclear formulations of the items.” – this sounds like the “experts” did a cursory job, perhaps not understanding their role.
Line 205. “As shown from the Average (SD) column no value falls below 3 (quite relevant), confirming that the experts evaluated the questionnaire as generally acceptable. “ – this could also be interpreted as experts wanting to appear cooperative and agreeable. (Journal reviewers rarely agree that a paper is perfect because they know their task is to identify problems.)
Line 210. “The majority of respondents (74.6%) were women with a high level of education (74%); approximately one-third (35.3%) of the sample was aged 50-69 years.” – the consumer sample was female, highly educated and older; does this reflect who buys sustainable foods? It does not represent average consumers. This convenience sample is noted in the Discussion. It is major problem with this paper.
Line 300. “The most important aspect of this research is that, to the best of the authors’ knowledge, this is the first study in which the development and testing of a web questionnaire to evaluate the consumers' attitude towards an emerging topic such as sustainability and source of proteins alternative to meat was deeply described and discussed.” - that might be the case, but experts (not consumers) developed the initial questionnaires.
Line 327. “The pilot test carried out on a casual sample of consumers provided very relevant information on how to change the questionnaire. This is an important aspect to consider be- cause the perspectives and the interpretation of the survey respondents (in this case the consumers) are different from those coming from the experts. “ – yes, but you started with experts designing the initial questionnaires. Did consumers have the opportunity to undo earlier errors?
Line 384. The limitations section is two sentences. This is a good start but needs to be expanded, including elaboration on the issues noted (convenience sample, selection bias). Also there is no discussion of any problems translating the original questionnaire into Italian.
Author Response
We thank the reviewer for his/her comments and suggestions. We addressed all the aspects that needed improvement in the attachment.
Hereafter are the answers point by point to the revisions of the manuscript as well as the responses to all the reviewers’ comments. We refer to the corrections to the line numbers of the revised manuscript. The Authors’ changes and the language revisions are shown in blue, and the eliminations are reported with the text strikethrough. Any modification can be easily viewed by the editors and reviewers.
RW1: p. 1. The first sentence of the Introduction does not make clear the place of health. Health is included by the FAO (first sentence but not after that)
AUTHORS: Acknowledging the point we reworded the first two sentences of the manuscript as follows eliminating the reference n.1 (see lines 31-42):
According to the Food and Agriculture Organization (FAO) and the World Health Organization (WHO), Sustainable Healthy Diets are dietary patterns that promote all dimensions of individuals’ health and wellbeing; have low environmental pressure and impact; are accessible, affordable, safe and equitable, and are culturally acceptable [1]. In recent years, several studies highlighted the negative impact that global food production has on water consumption, land use, and gas production. Considering the detrimental impact of current food systems, and the concerns raised about their sustainability, there is the need to promote diets that are protective of human health and the environment.
RW1: p. 2 line 45. Health is emphasized in this paragraph, but then it disappears beyond that.
AUTHORS: Considering your point, we rephrased as follows:
Shift towards more sustainable food patterns may be relevant not only for the environment perspective but also for public health consequences[6] [7] [8]. (See lines 49-50)
RW1: p. 3. The goals of this paper are important: validating national application from an international start. We often see the application of a questionnaire but not often the development of a questionnaire for national usage.
AUTHORS: Thanks you for this comment, actually this is exactly what the authors would like to transmit with the present paper.
RW1: Line 117. Instrument translation. How was this done? With a dictionary? With language experts? With consumers? If only done by experts it could be questioned from a linguistic perspective.
AUTHORS: The original language of the questionnaire was English (see annex of Ref n. 18). The items were simple, short, and written in language familiar to the target respondents considering that it was developed by a consumers’ organization and addressed to consumers in different EU countries including Italy. We considered not necessary the backward translation.
We added a short paragraph explaining the procedure, see lines 165-173:
In the first step of phase 1, the original questionnaire of the European Consumer Organisation [18] was translated into Italian. The translation was carried out by the authors of the present paper that were aware of the concepts that the questionnaire intended to measure and would provide a translation as much as possible closest to the original instrument [33]. No difficulties in the translation were encountered since the original questionnaire included items written in simple, short, and familiar language adapted to be disseminated among consumers. Considering the absence of misunderstandings or unclear wordings in the initial translations, the back-translation of the questionnaire into the original language was not carried out.
RW1: Sections 2.1.1 and 2.1.2. I question the sole use of experts in these roles, with no input by consumers.
AUTHORS: We got your point, however the validation protocol was designed according to Rodrigues et al. [35], Caruso et al. [36], and Yusoff [33]. Reporting two initial phases with experts’ involvement followed by the pilot that is the phase in which the questionnaire is tested on the assessment’s target group, the consumers in the present study. As mentioned in lines 484-485 of the discussion chapter, the content validity evaluation that used expert advice is a subjective procedure, and every expert could not represent all the dimensions of the content domain and, we assume that, also an initial assessment on the target groups (consumers) would be subjective and with limitations. The combinations of the inputs of the experts at the initial stages and the advices of the consumers later would compensate each others. In addition to that the major changes in our validation process came from the consumers assessment probably because the experts’ considerations were in line with the thinking of the research team that develop the questionnaire. The philosophy of the literature on the evaluation process is to have either the experts and the target groups comments, with different, but both valuable inputs.
In order to be most convincing we improved the methodology chapter (paragraph 2.1) better explain the reasons of the procedure increasing the literature supporting the methodology (see lines 148-157).
The questionnaire validation protocol was designed following a procedure largely reported in the literature [25–28], applied in different settings, and assessing different topics and constructs. The procedure could be summarized as follow: the initial analysis for the identification of the domains of interest was normally carried out by the team that developed the questionnaire [29], then the validity of the items was assessed in consecutive stages by requesting individual experts to evaluate the content relevance and simplicity of individual items and the entire set of items (questionnaire) as a tool [30–32]. Finally, the questionnaire was pilot tested in a sample of the target group. Hence, the stepwise procedure of the validation process has been divided into 3 phases (Figure 1):
RW1: Lines 181-186 – please check your English
AUTHORS: Thanks for the observations. Hereafter the revised sentences (see lines 247-254):
- A question on meat substitute (Q8) from Schösler et al. [22]
- A question on the role of meat in the diet (Q10) from Tarrega et al. [23]
- A question on consumers’ meat consumption attitude after changing production methods with increasing prices (Q11) and a question on different activities’ contribution to climate changes (Q14) from Malek et al. [24]
RW1: Line 187. “In the formal review of the content carried out by the first group of experts, comments were mainly related to redundancies and unclear formulations of the items.” – this sounds like the “experts” did a cursory job, perhaps not understanding their role.
AUTHORS: This could be an interpretation, however in this phase, we explained to the experts the meaning of the revisions and we did not have the impression that they do not understand their role either do not accurately perform their tasks. Actually as opposite interpretation we could say that the questionnaire was appropriately designed also considering that it becomes from another questionnaire already tested.
In order to accomplish with your comment, we better explained the procedure with the following reformulation of the sentence (see lines 255-265):
After the initial pool of the questionnaire, qualified experts reviewed the items. According to Tsang et al., [33], the purpose of this revision is to check for accuracy, absence of construction problems, and grammatical correctness. The formal review of the content was preceded by a thorough explanation of the role of the experts and the purpose of the revision. The experts were informed about the scope of the work and the importance to provide comments aimed to improve the quality of the assessment. Active and conscious participation of the experts was required. A general consensus of the experts on the relevance and validity of the questionnaire was reached with comments that were mainly related to redundancies and unclear formulations of the items.
RW1: Line 205. “As shown from the Average (SD) column no value falls below 3 (quite relevant), confirming that the experts evaluated the questionnaire as generally acceptable. “ – this could also be interpreted as experts wanting to appear cooperative and agreeable. (Journal reviewers rarely agree that a paper is perfect because they know their task is to identify problems.)
AUTHORS: As mentioned, the literature on questionnaire validation included the experts evaluation that even with limitations is generally accepted because it is coupled with the pilot carried out on the assessment’s target group. According to de Alwis et al. [30], although the power of the validation relies on the competencies of the experts that acted as raters, this methodology represents the best available for the validation of questionnaires. As further element of the reliability of the validation protocol is the fact that in our study the pilot test largely counterbalanced the expert acceptance of the questionnaire providing several elements of changing. Finally the role of experts in this procedure is different from that of the papers’ reviewers for journal publications; the expert reviewers in the case of validation protocol should, to the best of their ability, ensure that the items do not contain construction problems or are grammatically inaccurate. These concept are added to the manuscript at the lines 255-265 as reported in the previous answer.
RW1: Line 210. “The majority of respondents (74.6%) were women with a high level of education (74%); approximately one-third (35.3%) of the sample was aged 50-69 years.” – the consumer sample was female, highly educated and older; does this reflect who buys sustainable foods? It does not represent average consumers. This convenience sample is noted in the Discussion. It is major problem with this paper.
AUTHORS: Thanks for the comment. Our understanding of the pilot test in the validation protocol is that it should be carried out on the same population of the study respondents not necessarily representative of the population as the assessment must be; for these reasons we did not establish exclusion criteria as stated in the methodology. In our study the respondents were recruited with the so-called river sampling methodology a simple non-probabilistic approach frequently used to make claims about the general population in social science and policy research (see ref. n. 39).
In order to accomplish with your comments, these concepts were better elaborated with the sentences reported below and added in the methodology, results and discussion chapters.
We added the following sentence substantiated by a reference at lines 220-223 of the methodology chapter:
This simple non-probabilistic methodology to recruiting respondents online, by inviting them to follow a link to a survey placed on a web page, email, or other similar means, is defined as “river” sampling by Lehdonvirta et al., [39].
We added this sentence at the end of 3.3 paragraph of the results chapter (see lines 290-292):
No polarization of responses with scoring all near the bottom or near the top was observed since items were not eliminated for these reasons at this stage.
In addition to that we better elaborated the issue of selection bias of the pilot in the paragraph of limitations of the discussion chapter at lines 474-482. We reported the full improved paragraph of limitations later in the response to your specific comment.
RW1: Line 300. “The most important aspect of this research is that, to the best of the authors’ knowledge, this is the first study in which the development and testing of a web questionnaire to evaluate the consumers' attitude towards an emerging topic such as sustainability and source of proteins alternative to meat was deeply described and discussed.” - that might be the case, but experts (not consumers) developed the initial questionnaires.
AUTHORS: Yes the questionnaire was initially developed by experts and then modified according to the results of the pilot test on consumers using a largely codified procedure. Considering your comment, we better point out this aspect adding the following sentence in the discussion chapter at lines 379-381.
Using a procedure largely described in the literature [25-28] the questionnaire was initially developed by experts and then revised on the basis of the results of a pilot test on consumers.
RW1: Line 327. “The pilot test carried out on a casual sample of consumers provided very relevant information on how to change the questionnaire. This is an important aspect to consider be- cause the perspectives and the interpretation of the survey respondents (in this case the consumers) are different from those coming from the experts. “ – yes, but you started with experts designing the initial questionnaires. Did consumers have the opportunity to undo earlier errors?
AUTHORS: We completely agree on the importance of consumers’ perspective and interpretation of the survey. The pilot test was carried out anonymously, so a kind of restitution of results to respondents was not possible. However the further national assessment will be the occasion to share the revised version of the questionnaire and to discuss with a large audience the main findings.
RW1: Line 384. The limitations section is two sentences. This is a good start but needs to be expanded, including elaboration on the issues noted (convenience sample, selection bias). Also there is no discussion of any problems translating the original questionnaire into Italian.
AUTHORS: In consideration of your point we enriched the limitations paragraph in the discussion chapter (see lines 474-513):
This research has limitations and strengths. The main limitation is that the sample for the pilot is a convenience sample. In open web-based surveys, selection bias occurs inevitably [51] also for the effect of the “river sampling” that has the advantage of the facility to reach a large number of participants at limited costs, and the disadvantage of the selection bias. According to Lehdonvirta et al. [39], this kind of sampling can be used to describe certain phenomena and their safest use is in examining members of the sample itself avoiding inferences that could be affected by the selection biases. We performed the analysis of the pilot data in the light of these methodological inputs considering the pilot test an opportunity for the questionnaire developer to know if there is confusion about any items, and to explore any possible improvements of the items [33]. A limit of the present methodological study is that the content validity evaluation is a subjective procedure, and every expert could not represent all the dimensions of the content domain [27]. Another limitation is related to the fact that questions and response items were not randomized. The questionnaire was conceived by nutrition and public health experts that do not have specific competencies in environmental sustainability issues and also this could be considered a limitation of the research. However, in a modern approach nutritional recommendations should consider the environmental implications of food choices and the development of nutritional recommendations is a task of community nutrition specialists and public health experts. Our research has also strengths. The sample size used in the present study was in accordance with the classic rule of respondent-to-item ratios established by Kline [53] of using 2 to 20 subjects for each item of the questionnaire. Given the variation in the types of the questionnaire being used, there are no absolute rules for the sample size needed to validate a questionnaire [54]. However, we carried out a retrospective power test, calculating the error associated with the estimate of the average response for items from 1 to 10, assuming the maximum variance observed (10.5). A group of 150 guarantees a standard error of 0.26 (with a confidence interval amplitude of 1.05). Concerning the factor analysis there are several studies in the field that require a range of 3 to 10 subjects for each variable [55–57]. Other studies, however, state that the minimum sample size depends on the characteristics of the study, in particular on the level of communality of the variables and the level of factor over determination [58–60]. In our case this is an exploratory factor analysis and the ratio between the number of subjects (150) and the number of variables (21) is 7, within the indicated ranges. We have fairly high commonalities for most items and the first 4 factors explain 90% of the variability. Furthermore, we did not have any convergence problems. It is worthless to say that a larger samples are always better than smaller samples and that it is recommended that investigators utilize as large a sample size as possible. However, it is difficult to have large sample size in pilot validations study.
As far as concerning the translation issue, we added a paragraph at see lines 165-173 of the methodology chapter, as previously mentioned:
Thank you for the time and effort in reviewing the manuscript.

Reviewer 2 Report
Consumers’ attitudes towards sustainability
The manuscript requires copy editing by someone with English as a first language. There are numerous syntax and grammatical errors throughout, but it is not a reviewer’s role to fix these.
The topic would be of interest to readers, but the scope of the reported study is limited. It reports preliminary stages of validation on a very biased and small sample. No criterion validity is reported. Whilst the overall aims are stated and reasonable (Validation), it is not clear what is the purpose or future of the modified scale. If it is to improve the pan-European original it is based upon, then why is it tested in only one culture? Is it only for usage in Italy? If the latter, I’m not sure of it’s value in an international journal. If the overall aim (as in the research questions) is to demonstrate that questionnaires need validation, then this is not novel as this is long established in behavioural sciences. Validation may be novel to readers of Foods? Or, was the aim to improve the Italian dietary guidelines (L78)? At present there is no clarity.
Specific comments
Abstract L22 (and throughout).’ Response options’ or ‘responses’. These then become items in the new scale.
L26 I do not understand what “…. general, and not codified”. means?
L66 cultural capacity requires definition
L67-69 reports previous work on aspects of sustainability, but no implications of this previous work are stated. This is an isolated sentence that does not offer the reader any explanation. Are the authors trying to say that a more comprehensive approach is required? It seems improbable that only three studies have investigated sustainability.
L74 I believe other new dietary guidelines consider environmental impacts. Please reference.
L118 Was validity testing of the original pan-European questionnaire undertaken? State how this questionnaire was created.
L119 State the process of translation (back-translations?). What was the original language? Did every item translate into Italian without difficulty or require modification?
L157 The pilot study sample is very small and a major limitation. Were any incentives provided? The sample (female and highly educated) was very biased. Expand the limitations and implications (L386). The reference (45, L393) to sample size adequacy does not appear to account for the number of items for the factor analysis. Please review.
Was ethical approval obtained?
L179 Who made and how was the decision made to include 4 questions on meat and alternative proteins? Looking at the items, I am puzzled that participants were asked to consider alternative proteins ‘in the future’. In many jurisdictions these are available now and are extremely popular. I find Q2 responses particularly puzzling, The literature on sustainability, including, the present authors’ introduction, includes non-renewable inputs; fossil fuel and fertilizer usage, water usage, soil quality, soil loss, natural habitat protection, natural fauna and flora protection, lower carbon emissions, circular economy etc. and yet only one item (low environmental impact, a composite measure) alludes to any of these. All the other items seem to be halo effects, associations or expectations. It begs the question, how were these chosen originally? I note that the expert review panel had “competencies in nutrition and public health” (L131-132) but not in sustainability or the environment. This seems to be a further limitation and may account for missing items.
Section 3.3. State if the questions and response items were randomized. This may be particularly important when participants were asked to state three out of a range of options (e.g. Q2). There is usually a first order effect. If not, then state this as a limitation.
L230-234 This section seems to be misplaced. Was this format presented to the 150 or was this the result post-test (final version)?
Section 3.3.1 The first paragraph in this section appears to be quantitative (missing data) not qualitative. There is no indication that the items were tested for literacy levels. Looking at the original and modified version, some of the responses are constructed with complex language, including contingencies on the GMO items. Given these were only tested on highly educated participants, what are the implications for a more representative sample with lower literacy?
Section 3.3.2 and Table 2. Q6 and Q13 are also low and questionable. A Cronbach’s Alpha (CA) of 0.9 suggests some redundancy. How were these related to the number of items (response options)? All CAs suggest all Likert scale questions required some modification. Furthermore, it is not clear how the responses were treated. Were some responses reversed? - given that there appear to be positive and negative items?
L253. If the first four factors explain 90% of the variance, why are the other items included in the new scale? It is often the objective of scale development to derive a scale that is parsimonious. The final scale’s 71 items would be a burden on participants. This may relate to the ambiguity around the scale’s purpose. If the purpose is to measure succinctly consumers attitudes, then the shorter the better if it can account for 90% variance
There is no indication of further work or specific application. There is a need for criterion validity testing. Testing against behaviour is particularly important, as the attitude – behaviour gap is well documented in the environmental domain. Convergent validity with similar scales (e.g. those alluded to in the introduction) would also be helpful.
Author Response
We thank the reviewer for his/her comments and suggestions. We addressed all the aspects that needed improvement. in the attachment.
Hereafter are the answers point by point to the revisions of the manuscript as well as the responses to all the reviewers’ comments. We refer to the corrections to the line numbers of the revised manuscript. The Authors’ changes and the language revisions are shown in blue, and the eliminations are reported with the text strikethrough. Any modification can be easily viewed by the editors and reviewers.
RW2: The manuscript requires copy editing by someone with English as a first language. There are numerous syntax and grammatical errors throughout, but it is not a reviewer’s role to fix these.
AUTHORS: The full manuscript was deeply revised as far as concerning the language. Syntax and grammar errors were addressed and corrected.
RW2: The topic would be of interest to readers, but the scope of the reported study is limited. It reports preliminary stages of validation on a very biased and small sample. No criterion validity is reported. Whilst the overall aims are stated and reasonable (Validation), it is not clear what is the purpose or future of the modified scale. If it is to improve the pan-European original it is based upon, then why is it tested in only one culture? Is it only for usage in Italy? If the latter, I’m not sure of it’s value in an international journal. If the overall aim (as in the research questions) is to demonstrate that questionnaires need validation, then this is not novel as this is long established in behavioural sciences. Validation may be novel to readers of Foods? Or, was the aim to improve the Italian dietary guidelines (L78)? At present there is no clarity.
AUTHORS: Thanks for these comments that permitted us to better focus the objective of the study and of the paper. We answer further on the issue of the sample size and selection of respondents. We added a final paragraph on future use of the questionnaire and the implications of the present work. We confirm that we would like to improve the methodology of the original questionnaire that was developed without any validation procedure (see specific answer below). As for your point on validation, surely it is well known the importance to validate questionnaire, however, at far as we know, few papers reported in details the process of validation and its protocol in a way that could be easily replicated. It was not our intention to demonstrate that there is a need of validation (as you said, a well-known concept); however with this study we would point out that the validation process strongly affects the structure of the questionnaire implying several changes and adaptations. In consideration of these issues we retained important to share these info and the research to the readers of Foods journal (and similar) for whom the details could be relevant. Furthermore, we confirm that the validated questionnaire will be used for a nationally-representative assessment in Italy and potentially in other countries, probably with further fine tunings. Sorry to say that we disagreed with the point that the assessment in Italy would represent a limitation for publication in an international journal as Foods, especially considering that the Italian survey will be preceded with a solid methodological approach. Several papers in different journals and of various publishers described country specific studies that in some cases have also a very limited sample size (Hanning et al., 2009; Plichta & Jezewska-Zychowicz, 2019: Wiatrowski et al., 2021; Tsartsou et al., 2021; Annunziata & Mariani, 2021; Mazocco et al., 2018).
In order to better specify the scope of the present paper we added these sentences in the objectives paragraph of introduction section (see lines 132-137)
Specifically, the questionnaire resulting from the validation process will be aimed to evaluate consumers’ attitudes towards the themes of sustainability with a consolidated scientific methodology reinforcing and providing value to the findings of future assessments. A methodological improvement of the originally developed questionnaire [18] will be carried out and the extent of the changes after the validation process will be shown.
Specific comments
RW2: Abstract L22 (and throughout).’ Response options’ or ‘responses’. These then become items in the new scale.
AUTHORS: Thanks for the point. We replaced the term “options” with “response options” as suggested
RW2: L26 I do not understand what “…. general, and not codified”. means?
AUTHORS: In consideration of the observation we reworded the sentence (see lines 25-27):
This study showed the importance of the validation before the administration on a large scale of a questionnaire on a topic such as sustainability still lacking large support from consensus documents.
RW2: L66 cultural capacity requires definition
AUTHORS: Acknowledging the comment we rephrased the sentence (see lines 67-72):
These choices, however, are often conditioned by the level of knowledge, as resulting from a study carried out by Peschel et al. [10], further confirmedby Hartmann et al., (2021), reporting that the consumers’ knowledge is an important driver of motivation for appropriate food choices and habits environmentally protective [11].
RW2: L67-69 reports previous work on aspects of sustainability, but no implications of this previous work are stated. This is an isolated sentence that does not offer the reader any explanation. Are the authors trying to say that a more comprehensive approach is required? It seems improbable that only three studies have investigated sustainability.
AUTHORS: Considering your comment we reworded the sentence and we better linked the content with the rest of the paragraph. Please see lines 73-80:
Consumer attitude toward sustainability was studied, among others, with tools analyzing food waste and diet quality [12], knowledge of the environmental impact of food [11], and consumer literacy [13]. Despite the increasing evidence of the huge impact on the ecosystems of consumers’ dietary habits, most people are not aware of the environmental effects of food production and consumption[11]. In addition to that, the sustainability of diet is an aspect still not completely exploited, in which the risk of bias and “personal” interpretation is possible also in consideration of the limited sources of information based on consensus documents.
RW2: L74 I believe other new dietary guidelines consider environmental impacts. Please reference.
AUTHORS: Yes, you are right, the sentence was to much simple, we better articulated the concept adding the following sentences supported by bibliography (see lines 81-107):
The composition of diets and the quality of foods have direct effects on human health, a well-known and consolidated concept. However the indirect health effects caused by environmental changes associated with the processes of producing foods are less recognized. National dietary guidelines aim to provide advice for constructing healthy diets, thus the guidelines should consider both direct and indirect health consequences of the nutritional recommendations, including the environmental implications of food choices [14]. In the light of this commitment, several countries have started to incorporate sustainability considerations into their food policies and consumer education programs. Given the policy and programmatic implications of dietary guidelines, the development and integration of recommendations that promote specific food practices and choices would represent an obvious strategy for addressing sustainability, mainly in its nutrition and environmental dimensions. Such recommendations include for example: having a mostly plant-based diet, focusing on seasonal and local foods, reduction of food waste, consumption of fish from sustainable stocks ,and reduction of red and processed meat, highly-processed foods, and sugar-sweetened beverages [15]. At the European Level guidance on sustainability were mainly provided in term of recommendations for the selection of local seasonal products and reducing waste; however frequently these advices did not have an explicit mention of environmental aspects [16]. In the newly released Italian Dietary Guidelines [17] practical suggestions aimed to improve consumer’s behaviors in terms of environmental protection were provided in addition to health-promoting recommendations. One of the limitations of the Italian approach of the development of sustainability recommendations was that it was carried out without an evaluation of Italian consumers' awareness of sustainability.
RW2: L118 Was validity testing of the original pan-European questionnaire undertaken? State how this questionnaire was created.
RW2: L119 State the process of translation (back-translations?). What was the original language? Did every item translate into Italian without difficulty or require modification?
AUTHORS: We merged the answers to these two observations. The original pan-European questionnaire was not validated since the reasons of the work reported in this paper in which we developed an instrument that get inspiration from that already used by the BEUC but with a scientific and replicable methodology. We better pointed out this aspect with the sentences reported in lines 173-179).
The original language of the questionnaire was English (see annex of Ref n. 18). The items were simple, short, and written in language familiar to the target respondents considering that it was developed by a consumers’ organization and addressed to consumers in different EU countries including Italy. We considered not necessary the backward translation. We added a short paragraph explaining the procedure, see lines 166-173:
In the first step of phase 1, the original questionnaire of the European Consumer Organisation [18] was translated into Italian. The translation was carried out by the authors of the present paper that were aware of the concepts that the questionnaire intended to measure and would provide a translation as much as possible closest to the original instrument [33]. No difficulties in the translation were encountered since the original questionnaire included items written in simple, short, and familiar language adapted to be disseminated among consumers. Considering the absence of misunderstandings or unclear wordings in the initial translations, the back-translation of the questionnaire into the original language was not carried out. The original questionnaire [18] was created to better understand attitudes towards sustainable food consumption and the extent to which consumers realize the impact that their food choices have on the environment. The obstacles faced by consumers in adopting more sustainable food habits, and the measures they think are needed to make the sustainable choice easier were included in the assessment. In addition to that, for Italian consumers, it was considered relevant to assess the attitude towards sources of proteins alternative to meat. To this scope, a narrative review of the scientific literature of the most common repositories (e.g., Pubmed, Google Scholar, etc.) was carried out to search for questionnaires measuring the attitude of consumers towards the consumption of meat and the alternative sources of proteins.
RW2: L157 The pilot study sample is very small and a major limitation. Were any incentives provided? The sample (female and highly educated) was very biased. Expand the limitations and implications (L386). The reference (45, L393) to sample size adequacy does not appear to account for the number of items for the factor analysis. Please review.
AUTHORS: Thanks for the comment. In Italy, incentives for research study participation could not be provided by a Public Research Body. For these reasons often this kind of assessments were carried out with the help of consumers’ research agencies for sampling selection and recruitment of participants. As far as concerning the bias of the sample, our interpretation of the pilot was that it should be carried out on the same population of the study’s respondents but it should not be necessarily representative of the population as the assessment must be and did not require exclusion criteria as stated in the methodology. We conducted the pilot test of the questionnaire on the target respondents that were recruited with the so called river sampling methodology (Lehdonvirta et al., 2021), a simple non-probabilistic approach now frequently used to make claims about the general population in social science and policy research.
Concerning the adequacy of the sample size there is no recommendations on what numerosity is appropriate when conducting a factor analysis. The minimum suggested sample size varies from 3 to 20 the number of variables and absolute intervals should be from 100 to over 1000. The few studies on this topic rely on little empirical evidence to support these recommendations, with simulation studies having sample sizes of 180 individuals in different population conditions (Mundfrom et al., 2009). It is true that our sample size is not large but we based on several studies that require a range of 3 to 10 subjects for each variable [55–57]. Other studies, however, state that the minimum sample size depends on the characteristics of the study, in particular on the level of communality of the variables and the level of factor over determination [58–60]. In our case we performed an exploratory factor analysis and the ratio between the number of subjects (150) and the number of variables (21) is 7, within the indicated ranges. We have fairly high commonalities for most items and the first 4 factors explain 90% of the variability. Furthermore, we have not had any convergence problems.
In order to accomplish with your comments, these concepts were better elaborated with the sentences reported below and added in the methodology, results and discussion chapters.
We added the following sentence substantiated by a reference at lines 220-223 of the methodology chapter:
This simple non-probabilistic methodology to recruiting respondents online, by inviting them to follow a link to a survey placed on a web page, email, or other similar means, is defined as “river” sampling by Lehdonvirta et al., [39].
We added this sentence at the end of 3.3 paragraph of the results chapter (see lines 290-292):
No polarization of responses with scoring all near the bottom or near the top was observed since items were not eliminated for these reasons at this stage.
In addition to that we better elaborated the issue of selection bias and sample size of the pilot in the discussion chapter expanding the limitations section as required (see lines 474-513).
This research has limitations and strengths. The main limitation is that the sample for the pilot is a convenience sample. In open web-based surveys, selection bias occurs inevitably [51] also for the effect of the “river sampling” that has the advantage of the facility to reach a large number of participants at limited costs, and the disadvantage of the selection bias. According to Lehdonvirta et al. [39], this kind of sampling can be used to describe certain phenomena and their safest use is in examining members of the sample itself avoiding inferences that could be affected by the selection biases. We performed the analysis of the pilot data in the light of these methodological inputs considering the pilot test an opportunity for the questionnaire developer to know if there is confusion about any items, and to explore any possible improvements of the items [33]. A limit of the present methodological study is that the content validity evaluation is a subjective procedure, and every expert could not represent all the dimensions of the content domain [27]. Another limitation is related to the fact that questions and response items were not randomized. The questionnaire was conceived by nutrition and public health experts that do not have specific competencies in environmental sustainability issues and also this could be considered a limitation of the research. However, in a modern approach nutritional recommendations should consider the environmental implications of food choices and the development of nutritional recommendations is a task of community nutrition specialists and public health experts. . Our research has also strengths. The sample size used in the present study was in accordance with the classic rule of respondent-to-item ratios established by Kline [53] of using 2 to 20 subjects for each item of the questionnaire. Given the variation in the types of the questionnaire being used, there are no absolute rules for the sample size needed to validate a questionnaire [54]. However, we carried out a retrospective power test, calculating the error associated with the estimate of the average response for items from 1 to 10, assuming the maximum variance observed (10.5). A group of 150 guarantees a standard error of 0.26 (with a confidence interval amplitude of 1.05). Concerning the factor analysis there are several studies in the field that require a range of 3 to 10 subjects for each variable [55–57]. Other studies, however, state that the minimum sample size depends on the characteristics of the study, in particular on the level of communality of the variables and the level of factor over determination [58–60]. In our case this is an exploratory factor analysis and the ratio between the number of subjects (150) and the number of variables (21) is 7, within the indicated ranges. We have fairly high commonalities for most items and the first 4 factors explain 90% of the variability. Furthermore, we did not hve any convergence problems. It is worthless to say that a larger samples are always better than smaller samples and that it is recommended that investigators utilize as large a sample size as possible. However, it is difficult to have large sample size in pilot validations study.
RW2: Was ethical approval obtained?
AUTHORS: As mentioned in the methodology chapter, the data collection followed the Italian law for this kind of studies, meaning the Italian Data Protection Law (Legislative Decree 101/2018) in line with European Commission General Data Protection Regulation (679/2016). In the screenshot reported below it is possible to see the first page of the questionnaire (Screenshot n.1) showing how these info were shared with participants. In the text in Italian, we mentioned:
- The reason of data collection and the finality related only to research study.
- The names of the people responsible for the research and their affiliations.
- A statement saying, “The answers will remain confidential and will be used exclusively for statistical purposes in compliance with current legislation on the confidentiality of personal data”.
- Information regarding the processing of personal data, in which anonymity is guaranteed, the freedom to withdraw from the survey by participants at any moment and the fact that the data will be used for publication in scientific journals.
In addition to this, our research is not considered either as a medical experimentation, or a direct intervention on human subjects with diet changes or formulated food administration. The methodology and the data collection are similar to the surveys on the population carried out by the National Institute of Statistics that do not require ethical clearance but simply a reference to Personal data protection. In addition to that CREA is part of the National Statistical System (SISTAN) and guarantees individual data protection (Sette et al., 2010).
RW2: L179 Who made and how was the decision made to include 4 questions on meat and alternative proteins? Looking at the items, I am puzzled that participants were asked to consider alternative proteins ‘in the future’. In many jurisdictions these are available now and are extremely popular.
I find Q2 responses particularly puzzling, The literature on sustainability, including, the present authors’ introduction, includes non-renewable inputs; fossil fuel and fertilizer usage, water usage, soil quality, soil loss, natural habitat protection, natural fauna and flora protection, lower carbon emissions, circular economy etc. and yet only one item (low environmental impact, a composite measure) alludes to any of these. All the other items seem to be halo effects, associations or expectations. It begs the question, how were these chosen originally? I note that the expert review panel had “competencies in nutrition and public health” (L131-132) but not in sustainability or the environment. This seems to be a further limitation and may account for missing items.
AUTHORS: The decision of inclusion of 4 questions on meat and alternative proteins was made by the authors that decided to tackle these items in consideration of the Italians’ Mediterranean attitude in which the main source of vegetable proteins are the legumes, often not considered as alternative to meat. This aspect was discussed in the as a final conclusion paragraph added following your suggestion (see lines 523-534) and reported as response to your further specific comments. The hypothesis underling this assumptions was that the Italian consumers’ would shown a conservative attitude towards new foods or foods far from the tradition such as insects. As you mentioned, many of the foods included in the questionnaire response options are normally used and are very popular in other contexts, but not in Italy. It is useful to remember one of the scope of the questionnaire that would be used “as benchmark for developing specific actionable recommendations considering also the limits of the inclusion of sustainability in nutritional recommendations” (added at lines 534-542, following your suggestions) and reported as response to your further specific comments. The development of a recommendations very far form the habits will be of difficult achieving considering that it is important to consider the social and practical aspects, cultural factors, and consolidated behaviours. These considerations justified also the fact that the panel had competencies in nutrition and public health and not specific competencies in sustainability and environment because presently public health nutrition documents are claimed to consider the environmental impact of food consumption. Following your advice we added a sentence in the limitations paragraph of the discussion chapter (see lines 485-491) and reported as response your previous specific comment.
RW2: Section 3.3. State if the questions and response items were randomized. This may be particularly important when participants were asked to state three out of a range of options (e.g. Q2). There is usually a first order effect. If not, then state this as a limitation.
AUTHORS: No, questions and responses were not randomized. Thanks to providing this comments that permitted us to enrich the limitations’ section and the results. See lines 290-292 and lines 485-486 of the manuscript and the answer to previous comments.
RW2: L230-234 This section seems to be misplaced. Was this format presented to the 150 or was this the result post-test (final version)?
AUTHORS: This was the result of the pilot test and was not presented in this format to the sample of 150 respondents. We clarify this in the sentence (see lines 309-313):
As a result of the pilot test the majority of the answers were converted into continuous scales from 1 (very disagree) to 10 (very agree), with the obligation to answer for each item. Further reformulations were carried out for questions Q3, Q5, and Q8, allowing a dichotomous answer (Yes/No) for each item instead of limiting the selection to 3 response options.
RW2: Section 3.3.1 The first paragraph in this section appears to be quantitative (missing data) not qualitative. There is no indication that the items were tested for literacy levels. Looking at the original and modified version, some of the responses are constructed with complex language, including contingencies on the GMO items. Given these were only tested on highly educated participants, what are the implications for a more representative sample with lower literacy?
AUTHORS: We replaced the title of the paragraph putting “General analysis” instead of “Qualitative Analysis” (see line 294) . In the pilot one fourth (26%) of respondents had low literacy, so the items were tested also in a group with low education, smaller than the group with high literacy, but present. Actually, the sentence that we added in the methodology chapter to answer to one of your previous comment, stating “the original questionnaire included items written in simple, short and familiar language adapted to be disseminated among consumers (see lines 169-171)” could be considered also in the light of this observation. In fact we need to consider that the original questionnaires that included the GMO items was conceived by an entity working on consumers’ policy as is the BEUC (Bureau Européen des Unions de Consommateurs) that is used to interact with the different sub-groups of the consumers, including the less educated.
RW2: Section 3.3.2 and Table 2. Q6 and Q13 are also low and questionable. A Cronbach’s Alpha (CA) of 0.9 suggests some redundancy. How were these related to the number of items (response options)? All CAs suggest all Likert scale questions required some modification. Furthermore, it is not clear how the responses were treated. Were some responses reversed? - given that there appear to be positive and negative items?
AUTHORS: Thanks for the comments. After calculating the Cronbach’s Alpha, we carefully re-evaluated the questionnaire and found no redundancy between the items. This value allowed us to justify a high level of internal consistency. In addition, we confirm that for Cronbach’s calculation we reversed the scale of inverse questions.
In consideration of this comment we have added the following sentence in paragraph 2.2. (see lines 231-233)
For Cronbach’s calculation, negative wording questions were reversed into a positive scale.
RW2: L253. If the first four factors explain 90% of the variance, why are the other items included in the new scale? It is often the objective of scale development to derive a scale that is parsimonious. The final scale’s 71 items would be a burden on participants. This may relate to the ambiguity around the scale’s purpose. If the purpose is to measure succinctly consumers attitudes, then the shorter the better if it can account for 90% variance
AUTHORS: Thanks for the comments. We carried out the factor analysis to validate the internal consistency of the blocks. With this analysis we identified items that are not consistent with the others having also indications on their possible alternative collocation. For this reason, we kept all factors with a proportion of variance explained > 3%. This choice allowed us, for example, to notice how on factor 7 two items with a low commonality were isolated. These items were completely reformulated because they were not consistent with any of other questions. Furthermore, the aim of our factor analysis was not to create a shorted questionnaire.
RW2: There is no indication of further work or specific application. There is a need for criterion validity testing. Testing against behaviour is particularly important, as the attitude – behaviour gap is well documented in the environmental domain. Convergent validity with similar scales (e.g. those alluded to in the introduction) would also be helpful.
AUTHORS: In consideration of this comments we added the following sentences as conclusive paragraph of the discussion chapter (see lines 523-542):
The validated questionnaire resulting from this study will be used for a national-representative survey aimed at evaluating the degree of Italian consumers’ knowledge and understanding of food sustainability and whether the alternative source of proteins of the new generation or not usual in the Italian food culture, could substitute animal proteins in the diet. Italy would represent an interesting case study in the European context in consideration of the fact that Italian cuisine and consumers' attitude is largely based on the Mediterranean Diet principle [61], a model that has gained fame and honor, being a model that combines the prevention of Non-Communicable Chronic Diseases, longevity, and health with consumers’ acceptability and sustainability of the productive systems [62]. However in the Mediterranean context, the main sources of vegetable proteins are represented by cereals and legumes [63], the latter often not consumed as an alternative to meat. Since it would be of particular relevance to assess the attitude of Italians towards new foods and sources of proteins alternative to meat. In addition to that, the findings of the nationwide assessment could be used as a benchmark for developing specific actionable recommendations considering the limits of the inclusion of sustainability in nutritional advice contributing to the maximization of the capacity of dietary guidelines in creating a healthy food environment. Evaluation of the impact of these recommendations should be carried out in consideration of the gap between real food choices and good intentions either in terms of healthy eating and sustainability.
Extra References cited in this reply letter:
- Hanning RM, Royall D, Toews JE, Blashill L, Wegener J, Driezen P. Web-based Food Behaviour Questionnaire: validation with grades six to eight students. Can J Diet Pract Res. 2009 Winter;70(4):172-8. doi: 10.3148/70.4.2009.172. PMID: 19958572.
- Plichta M, Jezewska-Zychowicz M. Eating behaviors, attitudes toward health and eating, and symptoms of orthorexia nervosa among students. Appetite. 2019 Jun 1;137:114-123. doi: 10.1016/j.appet.2019.02.022. Epub 2019 Mar 3. PMID: 30840875.
- Wiatrowski M, Czarniecka-Skubina E, Trafiałek J. Consumer Eating Behavior and Opinions about the Food Safety of Street Food in Poland. Nutrients. 2021 Feb 11;13(2):594. doi: 10.3390/nu13020594. PMID: 33670190; PMCID: PMC7916948.
- Annunziata A, Mariani A. Do Consumers Care about Nutrition and Health Claims? Some Evidence from Italy. Nutrients. 2019 Nov 11;11(11):2735. doi: 10.3390/nu11112735. PMID: 31718014; PMCID: PMC6893455.
- Mazocco L, Gonzalez MC, Barbosa-Silva TG, Chagas P. Sarcopenia in Brazilian rural and urban elderly women: Is there any difference? Nutrition. 2019 Feb;58:120-124. doi: 10.1016/j.nut.2018.06.017. Epub 2018 Jul 25. PMID: 30391690.
- Daniel J. Mundfrom, Dale G. Shaw & Tian Lu Ke (2005) Minimum Sample Size Recommendations for Conducting Factor Analyses, International Journal of Testing, 5:2, 159-168, DOI: 10.1207/s15327574ijt0502_4.
Thank you for the time and effort in reviewing the manuscript.

Round 2
Reviewer 1 Report
Thank you for the revision, and the detailed reply to review.